# Supporting Smokers in Difficult Settings: Suggestions for Better Education and Counseling in Cancer Centers in Jordan

**Feras I. Hawari** [1,2,†], **Minas A. Abu Alhalawa** [1,‡], **Rasha H. Alshraiedeh** [3,4,§], **Ahmad M. Al Nawaiseh** [4,5,‖], **Alia Khamis** [4,5], **Yasmeen I. Dodin** [1] **and Nour A. Obeidat** [1,*]

1   Cancer Control Office, King Hussein Cancer Center, Amman 11941, Jordan
2   Section of Pulmonary and Critical Care, Department of Internal Medicine, King Hussein Cancer Center, Amman 11941, Jordan
3   Faculty of Pharmacy, The University of Jordan, Amman 11972, Jordan
4   Volunteer Research Program at King Hussein Cancer Center, Amman11941, Jordan
5   Faculty of Medicine, The University of Jordan, Amman 11972, Jordan
*   Correspondence: nobeidat@khcc.jo; Tel.: +962-6-5300460 (ext. 2204); Fax: +962-6-5345567
†   Current address: Office of Minister of Health, Amman 11118, Jordan.
‡   Current address: 6040 Belpree Rd., apt B118, Amarillo, TX 79106, USA.
§   Current address: Cancer Control Office, King Hussein Cancer Center, Amman 11941, Jordan.
‖   Current address: Jordan University Hospital, Amman 11942, Jordan.

**Abstract:** Continued smoking in cancer patients is commonly observed in Jordan. In a country that exhibits some of the highest smoking rates globally, enhancing patient education regarding the value of smoking cessation for cancer care is vital. The objectives of our study were to describe sociodemographic and clinical factors associated with continued smoking in Jordanian smokers after a cancer diagnosis; to identify reasons for smoking and knowledge regarding smoking's impact on care; to examine in a multivariable manner the factors associated with continued smoking, and to accordingly generate patient counseling recommendations. An interviewer-administered survey using the Theoretical Domains Framework was employed. Among 350 subjects (mean age 51.0, median 52.7), approximately 38% of patients had quit or were in the process of quitting; 61.7% remained smokers. Substantial knowledge gaps with regard to the impact of continued smoking on cancer care were observed. Remaining a smoker after diagnosis was associated with being employed, not receiving chemotherapy or surgery, having lower confidence in quitting, and having a lower number of identified reasons for smoking. Interventions to promote cessation in Jordanian cancer patients who smoke should focus on enhancing patient awareness about the impact of smoking in cancer care and raising perceived self-efficacy to quit.

**Keywords:** tobacco use; smoking cessation; cancer patients; Jordan

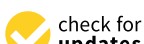



## 1. Introduction

The detrimental effects of continued smoking after a cancer diagnosis are well-documented and negatively impact treatment efficacy, tolerability to medications, side effects, quality of life, risk of recurrence or development of new primaries, and survival [1–3]. The value of smoking cessation in improving treatment outcomes and in lowering mortality rates in cancer survivors continues to be demonstrated in the literature [4–7] and is the reason why availing smoking cessation services (as well as appropriate training for healthcare providers) is ethically mandated in institutions providing cancer care [8–10].

Although broad guidance is provided with regards to how to address cancer patients who smoke [11–13], assisting cancer patients to quit is not a simple feat, and not all patients are able to quit [14–19]. While studies have been conducted to understand the perceptions of cancer patients who smoke in relation to quitting, these studies have largely focused on Western communities [20–28]. Conversely, there is limited evidence on the in-depth patient-reported experiences of Arab cancer patients who smoke, despite many Arab countries

such as Jordan reporting some of the highest smoking rates globally (the prevalence of tobacco use among adults in Jordan is 42%) [29]. The scarcity of studies is important to note, because both smoking and quitting experiences are multifaceted and vary across cultures and contexts. The lack of information limits the extent to which cancer care providers can understand and counsel patients living in challenging, smoking-burdened areas where tobacco use is a norm.

Our study aimed to examine changes to smoking behavior after a cancer diagnosis in a sample of Jordanian smokers receiving treatment in the largest Cancer Center in the country; and to assess which factors were associated with quitting in order to identify subgroups of cancer patients who would need more intensive focus when implementing smoking cessation interventions. We also sought to describe patients' tobacco-related beliefs and knowledge in the context of cancer care and generate specific recommendations about counseling Jordanian cancer patients. Ultimately, by providing content and evidence to guide counseling sessions, this study can enrich practitioners' knowledge with regards to what to anticipate with Jordanian cancer patients who smoke, and how to best support them and similar [Arab] cancer patients who smoke.

## 2. Materials and Methods

### 2.1. Setting and Sample

The study was conducted at King Hussein Cancer Center (KHCC). KHCC in Amman, Jordan is a Joint commission, disease-specific accredited comprehensive cancer center which offers cancer care to a substantial proportion of the Jordanian population, and also serves as a regional cancer treatment hub for the Middle East [30].

Patients in both in-patient and out-patient settings were screened for eligibility and informed consent was verbally elicited for the study (data collectors explained the purpose of the study using an Institutional Review Board (IRB) approved informed consent script, and verbal consent was then elicited; patients also were given a handout describing the key aspects explained in the informed consent process, and detailing the contact information of the Principle Investigator and IRB). Patients qualified for the study if they answered yes to the question "Are you currently a cigarette smoker, or were you a smoker up to the time of your diagnosis?" Those who reported they continued to smoke were considered current smokers while those who reported quitting were considered ex-smokers. Data collection began in July 2018, after piloting the tool and conducting the appropriate modifications and clarifications, and closed at the start of January 2020.

Using G-power software [31], we conducted power analyses across possible ranges of input parameters (we assumed small to moderate effect sizes). Calculations indicated that a sample size between 240 (for a moderate effect size) and 480 (for a small effect size) would be sufficiently powered to detect potential associations of interest. We, therefore, targeted a total of approximately 375 patients.

### 2.2. Questionnaire

An interviewer-administered Arabic cross-sectional survey was developed and reviewed for face, linguistic and content validity by two tobacco treatment dependence specialists. The survey was pilot-tested on five patients and further refined. The tool used as its framework is the Theoretical Domains Framework (TDF), which collates the elements of various behavioral change frameworks and presents a single, structured and comprehensive mechanism for studying the various factors that influence the performance of a behavior (e.g., smoking). Factors covered in the framework include knowledge, skills, social role and identity, beliefs about capabilities, optimism, beliefs about consequences, reinforcement, intentions, goal-setting, memory, attention and decision processes, environmental resources, social influences, emotion elicited through behavior, and behavioral regulation [32,33]. Within each of these factors, the literature was reviewed in order to ascertain the tools or approaches needed to measure each factor within smokers. Thus, various evidence-based tools and studies were used in the process of developing the final

survey (descriptions of each section are available in Table S1). Within reason, we attempted to comprehensively cover key aspects that surround the smoking and cessation process. Throughout the questionnaire, we also included open-ended probes to further explore aspects of the patient experience we may have missed in quantitative questions.

There were two versions of the questionnaire: an 'ex-smokers' version, and a 'current smokers' version. The questionnaires were largely similar with the exception of a few sections, where verb tenses in the questions differed. The final components of the tool are described in Table S1. Variables in the tool pertinent to the current analysis included:

- Tobacco use patterns (pre-diagnosis and current): daily cigarette consumption; days since the last cigarette; the number of quit attempts in the past; and smoking during treatment [34].
- Perceived nicotine dependence and risk of relapse [35,36].
- Reasons for smoking: both an open-ended question ("why did you/do you smoke?") and a series of 14 Likert scale items (strongly agree to strongly disagree) reflecting the common constructs driving cigarette use were employed [37–45]. Various reasons were listed, and a count of the number of reasons identified was generated.
- Smoking identity: five Likert scale items (strongly agree to strongly disagree) were included to capture the extent to which subjects perceived smoking was a part of their identity.
- Patient knowledge about the impact of smoking on various aspects of cancer treatment [46]. Various risks were listed, and a count of the number of cancer-specific risks identified was generated.
- Patient-reported receipt of information about the impact of smoking on cancer treatment.
- Importance and confidence in quitting smoking (or remaining quit) over the next 30 days, each measured on a ten-point scale [47].
- Clinical information: date of cancer diagnosis; cancer site and stage; treatments being received (chemotherapy, surgery, radiation, hormonal therapy, bone marrow transplantation); non-cancer comorbidities.
- Sociodemographic information such as age; gender; marital status; area of residence; and educational and employment statuses.

### 2.3. Operationalization of Key Smoking-Related Variables

Outcome variable: For the purpose of this analysis, we divided cancer patients into those who reported being current smokers, and those who reported quitting post-diagnosis (they identified as ex-smokers and had quit for a period of at least 30 days) or reported they were in the process of quitting (they identified as ex-smokers and had quit but had not yet reached a period of 30 days. We thus referred to them as 'in the process of quitting').

Reasons for smoking: With regards to the 14 Likert-scale measures capturing reasons for smoking, dichotomization was performed in order to present the proportions of patients agreeing with statements (versus disagreeing or having no opinion).

Identity related to smoking: With regards to the five Likert-scale measures capturing identity in the context of smoking, dichotomization was performed in order to present proportions of patients agreeing with statements (versus disagreeing or having no opinion).

Knowledge-related items: dichotomization was performed in order to present proportions of patients who thought smoking "definitely increases" or "definitely decreases" the seven listed risks during cancer treatment (versus those who were not able to definitively identify risks of smoking in the context of cancer care).

Receipt of information about the harms of smoking: this was a general open-ended question that was coded by the interviewer according to the listed risks in the knowledge-related items.

Risk of relapse: As detailed in Table S1, seven scored questions (capturing nicotine dependence and exposure to secondhand smoke) were used to calculate a final relapse score that could range from 1 to 13.

Importance and confidence in quitting smoking: a score ranging from 1 (not at all confident or not at all important) to 10 (extremely important or extremely confident) was used.

### 2.4. Statistical Analysis

A descriptive bivariate analysis was conducted to identify factors associated with being a cancer patient who remained a smoker post-diagnosis; perceived reasons for smoking and knowledge gaps also were evaluated to demonstrate the existing perceptions of cancer patients across the two groups (current smokers versus those who had quit or were in the process of quitting), with the aim of highlighting where counseling and patient support could be enhanced. Independent *t*-tests (Wilcoxon Rank Sum test when data were not normally distributed) and Chi-square tests were used for bivariate analyses of continuous and categorical variables, respectively.

A multivariable logistic regression was run to determine which variables were associated with being a current smoker. Key sociodemographic and clinical variables were included in the multivariable model, as were knowledge about smoking effects on cancer, and the perceived impact of smoking on the subject's health in general. Variables specifically included gender; age; region of residence in Jordan; marital status; educational status; employment status; stage; cancer type (solid tobacco-related, other solid tumors, hematological malignancies, other cancerous and precancerous diagnoses); receipt of chemotherapy for treatment; receipt of surgery for treatment; receipt of radiation for treatment; receipt of hormonal therapy for treatment; tobacco use pre-diagnosis; knowledge (number of cancer-specific risks identified); relapse score (Wisconsin relapse predicting score); a variable to capture the perceived impact of smoking on patients' health; the number of reasons identified by subjects with regards to why they smoke; having ever visited the smoking cessation clinic at the Center; time spent at the Center; and perceived confidence in and importance of quitting.

All statistical analyses were conducted using STATA 16 [48].

### 3. Results

#### 3.1. Descriptive Results

A total of 364 surveys were completed and 350 were included in the analytic sample (14 were dropped due to being largely incomplete). Descriptive characteristics of the sample are included in Table 1. Patients were approximately 51 years old, and the majority were males. Roughly 38% of 134 patients had quit smoking after their diagnosis. More specifically, 104 of the 134 had been smoke-free for at least 30 days, and 30 of the 134 were smoke-free but had not yet reached a 30-day smoke-free period. Current smokers were more likely to be employed, less likely to have visited the smoking cessation clinic at the Center, and were in the Center for shorter periods of time. Current smokers also reported lower scores of confidence in and importance of quitting. Clinical factors associated with remaining a smoker included not having received chemotherapy, radiation, or surgery. Notably, the Wisconsin Relapse predicting score was high, even among ex-smokers.

**Table 1.** Demographic, clinical, and tobacco use characteristics of a sample of cancer patients who currently smoke versus those who smoked up to diagnosis and then quit (column totals presented).

| Variable | Reported Quit or in the Process of Quitting, n (%) | Reported Current Smoking, n (%) | *p*-Value |
|---|---|---|---|
| **Demographics** | | | |
| **Age in years, mean (median)** | 51.1 (52.7) | 50.9 (52.7) | 0.93 |
| **Gender (proportion male)** | 117 (87.3%) | 178 (82.4%) | 0.22 |
| **Marital status** | | | |
| Single | 21 (15.9%) | 24 (11.4%) | |
| Married | 104 (78.8%) | 173 (82.0%) | 0.447 |
| Divorced/widowed | 7 (5.3%) | 14 (6.6%) | |
| **Education** | | | |
| Less than 12 years | 33 (25.6%) | 55 (25.6%) | |
| High school or diploma/vocational training | 46 (35.7%) | 93 (43.3%) | 0.284 |
| Bachelors or higher | 50 (38.8%) | 67 (31.2%) | |
| **Geographic residence** | | | |
| Central Jordan | 104 (78.8%) | 183 (84.7%) | |
| South Jordan | 7 (5.3%) | 7 (3.2%) | 0.165 |
| North Jordan | 17 (12.9%) | 25 (11.6%) | |
| Other areas | 4 (3.0%) | 1 (0.46 %) | |
| **Employment status \*** | | | |
| Working | 33 (25.6%) | 79 (38.9%) | 0.012 |
| Not working or retired | 96 (74.4%) | 124 (61.1%) | |
| Clinical characteristics | | | |
| **Months at cancer center: mean (median) \*\*** | 18.8 (7.5) | 16.1 (3.0) | 0.0006 |
| **Cancer site** | | | |
| Solid (respiratory) | 42 (32.1%) | 49 (22.4%) | |
| Solid (gastrointestinal, renal, urinary) | 25 (19.1%) | 54 (24.7%) | |
| Solid, other | 22 (16.8%) | 56 (25.6%) | 0.075 |
| Leukemias, lymphomas | 35 (26.7%) | 53 (24.2%) | |
| Other | 7 (5.3%) | 7 (3.2%) | |
| **Staging** | | | |
| Localized | 38 (29.0%) | 84 (38.4%) | |
| Regional | 32 (24.4%) | 44 (20.1%) | |
| Metastatic | 45 (34.4%) | 67 (30.6%) | 0.719 |
| Other staging | 11 (8.4%) | 18 (8.2%) | |
| Not applicable | 3 (2.3%) | 4 (1.8%) | |
| Unknown | 2 (1.5%) | 2 (0.9%) | |
| **Treatment** | | | |
| Chemotherapy received \* | 88 (67.2%) | 100 (45.7%) | 0.000 |
| Surgery received \* | 40 (30.5%) | 37 (16.9%) | 0.005 |

**Table 1.** *Cont.*

| Variable | Reported Quit or in the Process of Quitting, n (%) | Reported Current Smoking, n (%) | *p*-Value |
|---|---|---|---|
| Radiation received * | 39 (29.8%) | 30 (13.7%) | 0.000 |
| Hormonal therapy received | 5 (3.8%) | 12 (5.5%) | 0.440 |
| **Tobacco use characteristics** | | | |
| **Age of smoking initiation, mean(median)** | 17.3 (17) | 17.3 (17) | 0.99 |
| **Daily cigarettes pre-cancer, mean (median) *** | 30.9 (30) | 35.1 (30) | 0.05 |
| **First cigarette within a half hour (pre-cancer)** | 92 (70.2%) | 161 (73.5%) | 0.482 |
| **Daily cigarettes in the past month, mean (median) *** | 1.8 (0) | 20.8 (20) | 0.0000 |
| **Days smoked in past month, mean(median) *** | 2.5 (0) | 27.7 (30) | 0.0000 |
| **Ever visited smoking cessation clinic *** | 68 (51.9%) | 73 (33.3%) | 0.000 |
| **Wisconsin predicting relapse score, mean(median)** | 8.3 (8) | 8.6 (9) | 0.170 |
| **Importance of quitting, mean, on a scale of 1 to 10 (median) *** | 9.0 (10) | 8.1 (10) | 0.0008 |
| **Confidence in quitting, mean, on a scale of 1 to 10 (median) *** | 8.3 (10) | 6.4 (7) | 0.0000 |

* Significant differences (Chi-square, *t*-test) across groups, *p* < 0.05; ** Significant differences in Wilcoxin Rank-sum test across groups, *p* < 0.05.

### 3.2. Perceptions and Knowledge Related to Smoking

Perceptions pertaining to tobacco use are displayed in Table 2. With regards to motives for smoking, the least resonating motive for smoking was weight control. The most frequently identified motives for smoking (in at least 70% of each group) included negative reinforcement, boredom, positive reinforcement, smoking around others, and automaticity. Significantly more patients who had quit reported smoking more around people and enjoying the taste and sensorimotor experience of smoking. In more than half the total sample, smoking did not seem to be strongly linked to identity. Table 2 also includes data on knowledge regarding the harms of smoking during cancer treatment. Across all groups, at best no more than 49% of subjects were able to definitely identify the specific risks associated with smoking during cancer treatment.

In an open-ended question to shed light on why patients started smoking, 'habit' was cited most frequently (by 25.6%) amongst those who provided an answer, followed by 'starting due to being around smokers' (cited by 20.5%). Approximately 11% began smoking due to negative events or stress (results not displayed in tables).

Relatedly, when we probed the receipt of information from staff about the continued harms of smoking, 41.1% were told smoking would lower the efficacy of their treatment. However, of the 77 patients who had had surgery around the time of the survey, 5 (6.5%) reported being told that smoking would impact wound healing, 3 (3.8%) reported being told that smoking may cause post-surgical infections, and 2 (2.6%) reported being told that smoking could cause anesthesia complications. Of the 195 patients who took chemotherapy and/or radiation around the time of the survey, 92 (47.2%) reported being told that smoking could impact treatment efficacy; and 33 (16.9%) reported being told that smoking could increase treatment side effects. Only 24 patients in total (6.9%) reported being told about the increased risk of cancer recurrence with continued smoking, while 29 (8.3%) were told

about the increased risk of new cancers with continued smoking. Nevertheless, smoking was broached by staff: almost the entire sample (95%) was asked about their tobacco use, 80.6% were advised to quit smoking, 69.7% were told that continued smoking was harmful, and 81.4% were referred to the smoking cessation clinic (results not displayed in tables).

**Table 2.** Perceptions related to tobacco use among a sample of Jordanian cancer patients who currently smoke or smoked up to diagnosis and then quit (column totals presented).

| Variable | Reported Quit or in the Process of Quitting, n (%) | Reported Current Smoking, n (%) | *p*-Value |
|---|---|---|---|
| **Reasons underlying smoking** ψ | | | |
| Affiliative attachment (quitting like losing friend) *n* = 337 | 81 (62.7%) | 109 (52.6%) | 0.06 |
| Automaticity (reach for cigarette without realizing) *n* = 345 | 96 (73.8%) | 165 (76.7%) | 0.54 |
| Loss of control (weak in face of a cigarette) *n* = 346 | 91 (68.9%) | 141 (65.9%) | 0.56 |
| Cognitive enhancement (smoking helps me focus) *n* = 338 | 78 (60.5%) | 124 (59.3%) | 0.84 |
| Cues/goads (visual triggers make me want to smoke) *n* = 344 | 84 (63.6%) | 124 (58.5%) | 0.34 |
| Cues or goads (seeing smoker makes me want to smoke) *n* = 343 | 96 (73.3%) | 140 (66.0%) | 0.16 |
| Social (smoke more around people) * *n* = 342 | 106 (81.5%) | 144 (67.9%) | 0.006 |
| Taste (I like the taste of cigarettes) * *n* = 344 | 85 (65.4%) | 100 (46.7%) | 0.001 |
| Weight control (smoke to control weight) *n* = 324 | 16 (13.2%) | 30 (14.8%) | 0.70 |
| Negative reinforcement (smoke when angry) *n* = 347 | 117 (88.6%) | 1924 (89.3%) | 0.85 |
| Positive reinforcement (smoke when relaxed) *n* = 346 | 106 (80.3%) | 164 (76.6%) | 0.42 |
| Social (smoke to socialize) *n* = 333 | 74 (57.8%) | 117 (57.1%) | 0.89 |
| Sensorimotor (enjoy handling cigarette) * *n* = 339 | 80 (63.5%) | 112 (52.6%) | 0.05 |
| Smoke when bored *n* = 343 | 111 (84.1%) | 175 (82.9%) | 0.78 |
| Number of reasons for smoking that were identified | 9.6 | 9.1 | 0.08 |
| **Smoking as part of identity** ψ | | | |
| Cannot/could not imagine life without cigarette * (*n* = 344) | 74 (57.4%) | 93 (43.3%) | 0.01 |
| Others cannot/could not imagine you as nonsmoker * (*n* = 340) | 60 (46.9%) | 76 (35.9%) | 0.04 |
| Felt you were/would lose part of self when quitting (*n* = 339) | 46 (35.9%) | 60 (28.4%) | 0.15 |
| Smoking makes/made you special/distinct (*n* = 343) | 27 (20.8%) | 30 (14.1%) | 0.11 |
| Felt or feels like I'd/I'll never quit * (*n* = 343) | 76 (58.9%) | 67 (31.3%) | 0.000 |
| **Perceived effects of smoking on health in general and cancer in particular** | | | |
| Smoking impacts/impacted my health "a lot" * | 82 (61.2%) | 106 (49.1%) | 0.03 |
| Smoking "definitely" increases risk of recurrence * | 59 (44.0%) | 70 (32.4%) | 0.03 |
| Smoking "definitely" increases stress | 34 (25.4%) | 39 (18.1%) | 0.10 |
| Smoking "definitely" increases surgical complications | 51 (38.1%) | 70 (32.4%) | 0.28 |
| Smoking "definitely" lowers survival | 60 (44.8%) | 85 (39.4%) | 0.32 |
| Smoking "definitely" lowers chemotherapy/radiation efficacy | 48 (35.8%) | 68 (31.5%) | 0.40 |
| Smoking "definitely" increases chemotherapy/ radiation side effects | 41 (30.6%) | 61 (28.2%) | 0.64 |
| Smoking "definitely" increases pain level | 33 (24.6%) | 44 (20.4%) | 0.35 |
| Count of the number of cancer-specific risks identified (out of 7) ** | 3.0 | 2.5 | 0.03 |

* Significant differences (Chi-square, *t*-test) across groups, *p* < 0.05; ** Significant differences in Wilcoxin Rank-sum test across groups, *p* < 0.05; ψ Numbers (percentages) agreeing with statement are reported.

### 3.3. Multivariable Regression

Results of the multivariable logistic regression of the odds of being a current smoker (not making any conscious changes towards quitting) after diagnosis are displayed in Table 3. Patients reporting current employment had 2.38 times higher odds of being current smokers. Conversely, receipt of chemotherapy or undergoing surgery were associated with 62% and 69% lower odds of current smoking (respectively). Furthermore, for each unit increase in the confidence in quitting (or remaining quit) scale, the odds of being a current smoker decreased by 24%. Patients who perceived their smoking to have impacted their health in general also were 50% less likely to be current smokers. Finally, as the number of reasons for smoking [that patients identified] increased, the odds of being a current smoker decreased by 11%.

**Table 3.** Multivariable logistic regression predicting the odds of being a current smoker (relative to quitting or being in the process of quitting) across various patient-related factors.

| Variable | Odds Ratio (95% Conf. Interval) | *p*-Value |
|---|---|---|
| **Male (versus female)** | 0.54 (0.21–1.41) | 0.21 |
| Age | 0.99 (0.96–1.02) | 0.71 |
| **Region (Central)** | | |
| South | 0.88 (0.21–3.69) | 0.86 |
| North | 1.45 (0.63–3.35) | 0.38 |
| Other | 0.14 (0.01–1.64) | 0.12 |
| **Marital status (married)** | | |
| Divorced/widowed | 0.87 (0.24–3.12) | 0.83 |
| Single | 0.46 (0.16–1.36) | 0.16 |
| **Education (less than 12 years)** | | |
| High school/diploma | 1.52 (0.71–3.27) | 0.29 |
| Bachelors or more | 0.75 (0.33–1.69) | 0.49 |
| Currently working * | 2.38 (1.23–4.62) | 0.01 |
| **Stage (localized)** | | |
| Regional | 0.74 (0.32–1.71) | 0.49 |
| Metastatic | 1.16 (0.56–2.39) | 0.70 |
| Other staging | 1.33 (0.44–3.98) | 0.61 |
| **Primary site (respiratory)** | | |
| Solid (gastrointestinal, renal, bladder) | 1.64 (0.66–4.05) | 0.28 |
| Solid other | 1.49 (0.50–4.47) | 0.47 |
| Hematological malignancies | 1.01 (0.39–2.65) | 0.98 |
| Other | 0.39 (0.09–1.70) | 0.21 |
| **Daily cigarettes pre-diagnosis** | 1.02 (1.00–1.04) | 0.06 |
| **Wisconsin relapse predicting score** | 1.04 (0.87–1.25) | 0.65 |
| **Number of cancer-specific risks identified** | 1.11 (0.96–1.27) | 0.16 |
| **Felt smoking impacted their health *** | 0.50 (0.25–0.98) | 0.04 |
| **Reasons for smoking *** | 0.89 (0.80–0.99) | 0.04 |
| **Surgery therapy received *** | 0.31 (0.15–0.66) | 0.000 |
| **Radiation therapy received** | 0.47 (0.21–1.06) | 0.07 |
| **Hormone therapy received** | 0.99 (0.21–4.64) | 0.99 |

**Table 3.** *Cont.*

| Variable | Odds Ratio (95% Conf. Interval) | *p*-Value |
|---|---|---|
| **Chemotherapy received *** | 0.38 (0.19–0.77) | 0.01 |
| **Ever went to SCC** | 0.64 (0.35–1.15) | 0.13 |
| **Importance of quitting** | 0.93 (0.80–1.09) | 0.40 |
| **Confidence in quitting *** | 0.76 (0.68–0.86) | 0.000 |
| **Months at center** | 0.99 (0.98–1.00) | 0.08 |

* Significant *p*-value using cut-off of 0.05.

## 4. Discussion

Our study sought to shed light on an understudied group of patients—Jordanian cancer patients who smoke—in order to better understand the unique perspectives of this group and the factors that tend to be associated with remaining a current smoker after a cancer diagnosis. Detailed studies that highlight the counseling needs of Arab patients in this context are scarce, an unfortunate reality given that Arab countries such as Jordan report some of the highest rates of smoking globally [49]. It is notable that our sample of Jordanian cancer patients had a young median age (52.7 years), consistent with national data that confirm that Jordanian cancer patients are generally diagnosed at median ages that are relatively low (56 years of age, relative to the median age of 66 in patients in the US, for example) [50,51].

In the general literature, factors that have been associated with greater quit rates among cancer patients have included: smoking-related cancer sites, older age, being male, being in a higher socioeconomic status group, having high confidence to quit, having lower perceived difficulties in quitting, having higher cancer-related risk perceptions and knowledge, being anxious about cancer recurrence, using specific behavioral techniques to quit, not living with smokers, having smoked for shorter durations, and heaviness of smoking pre-diagnosis [18,20,52–58]. In our study, being currently employed was associated with greater odds of remaining a smoker. This observation could have more than one explanation: for example, the association could be the result of exposure to smoke at work, or experiencing stress-related work which drives continued smoking. We also did not observe a significant effect of having ever visited the smoking cessation clinic on being a quitter in our multivariable model.

Unlike other studies that demonstrated lower odds of quitting with more advanced stages [59,60], our results revealed that receiving treatment (chemotherapy or surgery) was the only clinical variable associated with lower odds of quitting. Receipt of treatments such as chemotherapy has been associated with quitting smoking in other studies [60].

Our findings are consistent with other studies that reveal knowledge gaps among cancer patients who smoke [61], and a lower perceived health impact of smoking on health among current smokers [62]. Our findings underscore the continuing gaps in tobacco-related knowledge among Jordanian cancer patients, and the pressing need for healthcare providers to address these gaps through customized and timely counseling. Several specific recommendations can be made:

- Subjects in our study simultaneously identified with various motives (reasons) for smoking, implying that various situational scenarios need to be discussed with patients during behavioral counseling; we also observed a significant inverse association between a number of reasons identified and the odds of being a current smoker, suggesting that perhaps current smokers downplayed the factors that prompted their smoking behaviors, while those who had quit or were trying to quit were more cognizant of the situations and reasons behind their smoking.
- The association of confidence in quitting with a lower odds of being a current smoker emphasizes the importance of working to build self-efficacy in Arab cancer patients who smoke.

- The finding that even among ex-smokers relapse predicting scores were high underscores the highly challenging conditions that smokers in countries such as Jordan present with, and the need for intensive counseling about the nature of relapse and how to preempt common factors associated with relapse.
- The association of employment with greater odds of remaining a smoker suggests that probing with patients their home and work environments and dedicating time to discuss their potential impact on the quitting experience could be of value. Providers may not realize the value of discussing such topics in an in-depth manner during counseling.
- Despite being asked about tobacco use and being provided with "Ask, Advise, Refer (AAR)", patient knowledge with regards to more detailed mechanisms of tobacco harm and how smoking impacted their cancer care was limited. This suggests poor patient-provider communications during the "Advise" phase, and it is likely that healthcare practitioners may not be providing sufficient details. It is also likely that patients are not processing or retaining the detailed information they are provided with. Such knowledge gaps reiterate the need for providers to repeatedly raise the issue of smoking and the value of cessation in the context of a patient's specific cancer treatment, and to ask patients to explain their own understanding of smoking cessation's role in their cancer care.

Our study has some limitations. We did not select a random sample of cancer patients who smoke, but data collectors were instructed to frequent all potential waiting areas and approach all patients to the best of their capacity, across a period of approximately 18 months. Our observed rate of 61.7% of current smoking may therefore not be a generalizable prevalence rate of continued smoking. Nevertheless, the percentage of current smokers is not unusual, and other studies have indicated that a substantial proportion of smokers continue to smoke after a diagnosis [18,63,64]. Older studies on cessation rates in Jordanian cancer patients also indicate that relapse rates are very high [16,65].

Our study generated needed data with regards to Jordanian cancer patients, but also reveals continuing information needs. For example, although we did not find a significant effect of visiting the smoking cessation clinic on being a quitter (in our multivariable model), in order to better decipher the impact of smoking cessation services, a more in-depth analysis of the frequency and nature of visits to the clinic would have been needed, which was beyond the scope of our current analysis. We also studied patients in a cross-sectional manner. Smoking cessation is a challenging and dynamic process, and following patients over time through longitudinal studies to better understand how the treatment and post-treatment journey changes smoking practices (and to what extent quitters maintain abstinence) would be of great benefit.

Our study is the first, to our knowledge, to study the perspectives of Jordanian cancer patients who smoke in an in-depth manner. The findings observed can be used to improve the counseling content that Jordanian cancer patients need to increase cessation attempts in this challenging patient group.

## 5. Conclusions

Continued smoking after a cancer diagnosis in Jordan is commonly observed. In a country that exhibits some of the highest smoking rates globally, enhancing patient education regarding the value of smoking cessation in cancer patients is vital. Factors such as perceptions regarding the ability to quit, smoker's environment, and cancer care experiences, and smoking's impact on these, are potential points to focus on to motivate cessation.

Interventions to promote cessation in Jordanian cancer patients who smoke should focus on enhancing patient awareness about the deleterious impact of smoking on cancer care and prognosis, and on raising perceived self-efficacy to quit. Practitioners need to continually emphasize to patients—regardless of cancer type—the impact of smoking on multiple facets of cancer care and survival, as knowledge gaps with regard to this area are evident.

**Supplementary Materials:** The following supporting information can be downloaded at: https://www.mdpi.com/article/10.3390/curroncol29120732/s1, Table S1: Overview of measured items.

**Author Contributions:** F.I.H. and N.A.O. contributed to the conceptualization and design of the study, questionnaire development, data collection, formal data analysis, results interpretation, and manuscript development. M.A.A.A. and A.M.A.N. contributed to the conceptualization and design of the study, data collection, and conducted the interviews. R.H.A., conducted the interviews, contributed to data collection, and manuscript revision. R.H.A., A.M.A.N. and A.K. conducted the interviews, and contributed to data collection. Y.I.D. contributed to data access and collection, formal analysis, and manuscript revision. All authors have read and agreed to the published version of the manuscript.

**Funding:** This research was supported by funds from the Intramural Research Grant Program at King Hussein Cancer Center.

**Institutional Review Board Statement:** The study was reviewed and approved by King Hussein Cancer Center (KHCC's) Institutional Review Board (IRB) (18KHCC03).

**Informed Consent Statement:** Written informed consent was waived and informed consent was verbally elicited for the study.

**Data Availability Statement:** Data cannot be shared publicly because of institutional regulations. Data requests are reviewed and approved by the Institutional Review Board at King Hussein Cancer Center (contact Linda Kateb, at IRBOffice@KHCC.JO). For researchers who meet the criteria for access to confidential data, data can then be shared.

**Acknowledgments:** The authors gratefully acknowledge the data collection efforts of Osama Bassah (medical student) and Murad Kharabsheh (resident physician).

**Conflicts of Interest:** The authors declare that they have no known conflict of interest or personal relationships that could have appeared to influence the work reported in this paper.

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
