# Peer review of "Supporting Smokers in Difficult Settings: Suggestions for Better Education and Counseling in Cancer Centers in Jordan"

_curroncol, doi:10.3390/curroncol29120732_

Round 1
Reviewer 1 Report
The authors of this report have conducted a large study of smoking cessation behaviors in cancer patients in Jordan. The study has some novel aspects to it including the findings from a Mid-Eastern population with very high smoking rates and some predictor variables that have for the most part not been examined in previous studies. Overall the paper is well written. There are some issues that should be addressed to strengthen the findings and interpretation.
1. Table 2 should have p-values for the comparison of quitters vs non-quitters.
2. Table 3 title should be more specific. I assume it is the predicting the odds of quitting.
3. One of the more striking findings is that cancer therapy was associated with lower quit rates. The authors should discuss this in the discussion section and offer an explanation for the finding. In particular, there were no differences in quitting by cancer stage, which is correlated with the receipt of cancer therapy which makes the finding especially notable. Why is stage not a predictor?
4. Since both in-patient and out-patient settings were screened, how did the authors make sure that some patients weren't interviewed twice?
5. Please specify how complete are the medical history records. Cancer treatment and referral patterns are complex, where depending on the cancer type and residence of the patient, some patients may have had surgery locally and then referred for follow-up care at the King Hussein Center. Were the authors able to account for this? If case-ascertainment was done by just reviewing the scheduling logs of clinics, such information on pre-King Hussein treatment might have been missed.
6. In-patient and out-patient subjects might be quite different in a number of aspects. It would be helpful to see if this is a predictor variable in Tables 1 and 3.
7. How did authors operationally define "process of quitting?" Was this self-report.
8. It is surprising that the median age is only 51. The median age in the U.S SEER program is 67, and even older for smoking-related cancers. For lung cancer it is about 70. Please double-check that this is reported correctly. If so, the authors should discuss why cancer age is so much younger in Jordan and the implications for smoking cessation in this population.
9. Section 2.4. Please specify the software and version used for the analysis. It's unclear what were the covariates in the logistic model. Please specify here and in a footnote of Table 3.
10. In the limitations section, authors should acknowledge that some if not many patients who reported quitting may relapse.
Reviewer 2 Report
The paper addresses a socially relevant topic, tobacco use as an important risk factor associated with cancer diagnosis, however, it is important that the following aspects of the paper be reviewed:
-The objective is not clear, different objectives were identified
-The objective is not in the abstract of the paper
-In the section "operationalization of variables" it is not understood, and the objective of the section is to clearly define the study variables.
- The results have no order and are not presented according to the objective of the study
- The tables are not clear and have a lot of information, attention must be paid to the format, in some tables information on percentages and frequency is included but it is not indicated what it refers to in the first line of the same.
- The discussion should be reviewed and an analysis of the results of the study should be made with respect to other studies and proposals for future research should be made.
Round 2
Reviewer 1 Report
Authors have satisfactorily addressed the comments. The paper is much improved.